# Evaluating Bayesian deep learning for radio galaxy classification

**Devina Mohan**[1]                    **Anna M. M. Scaife**[1, 2]

[1]Jodrell Bank Centre for Astrophysics, Department of Physics & Astronomy, University of Manchester, UK
[2]The Alan Turing Institute, London, UK

## Abstract

The radio astronomy community is rapidly adopting deep learning techniques to deal with the huge data volumes expected from the next generation of radio observatories. Bayesian neural networks (BNNs) provide a principled way to model uncertainty in the predictions made by such deep learning models and will play an important role in extracting well-calibrated uncertainty estimates on their outputs. In this work, we evaluate the performance of different BNNs against the following criteria: predictive performance, uncertainty calibration and distribution-shift detection for the radio galaxy classification problem.

## 1   INTRODUCTION

Bayesian neural networks (BNNs) have tremendous potential in scientific applications of machine learning. However, most large scale evaluations of BNNs focus on well-curated terrestrial datasets with lots of labelled examples [Wilson et al., 2022, Vadera et al., 2022]. In contrast, in radio astronomy, the largest labelled datasets are of the order $10^3$ [Porter and Scaife, 2023]. In this work we present an evaluation of Bayesian deep learning for radio astronomy, using the morphological classification of radio galaxies as a benchmark. Supervised CNNs have been the most widely used solution to this problem since their introduction to the field by Aniyan and Thorat [2017]. That work adopted the canonical morphological division of radio galaxies into Fanaroff-Riley Type I (FRI) and Type II (FRII), which has persisted as the most common classification scheme for radio galaxies in the literature for more than 40 years [Fanaroff and Riley, 1974]. More recently the FR classification scheme has been used to demonstrate improvements in efficiency and accuracy for a variety of deep-learning models within both the supervised [Lukic et al., 2019, Becker et al., 1995, Bowles et al., 2021,

Scaife and Porter, 2021] and unsupervised [Slijepcevic et al., 2022] learning regimes.

With recent improvements in the sensitivity and resolution of modern radio astronomy observatories, the morphological detail recovered in images of radio galaxies has indicated that more complex relationships exist beyond the original FR dichotomy [Mingo et al., 2019]. Whilst a more nuanced analysis will certainly be enabled by the development of increasingly fine-grained automated classification, the underlying continuum of physical processes that are represented by this diversity of morphology is perhaps better captured by understanding the confidence with which certain galaxies are assigned to different labels by these models. However, the confidence of individual predictions is not necessarily reflected in standard metrics for deep learning, but instead requires models to focus on uncertainty quantification of model predictions rather than raw performance [Mohan et al., 2022].

BNNs provide a principled way to model uncertainty [MacKay, 1992a,b] by specifying priors, $P(\theta)$, over the neural network parameters, $\theta$, and learning the posterior distribution, $P(\theta|D)$, over those parameters, where $D$ is the data. Recovering this posterior distribution directly is intractable for neural networks. Several techniques have been developed to approximate Bayesian inference for neural networks. We consider Hamiltonian Monte Carlo [HMC; Neal and Hinton, 1998, Neal et al., 2011], Variational Inference [VI; Blei et al., 2016, Blundell et al., 2015, Graves, 2011], last-layer Laplace approximation[LLA; Daxberger et al., 2021], MC Dropout [Gal and Ghahramani, 2015] and Deep Ensembles [Lakshminarayanan et al., 2017] for our application. We focus our evaluation on the following criteria: predictive accuracy, uncertainty calibration and ability to detect different types of distribution shifts.

In Section 2 we give a brief overview of the BNNs considered in this work; in Section 3 we describe the datasets used to train and evaluate our BNNs; in Section 4 we describe our experimental setup and finally in Section 5 we present

our evaluation, followed by a discussion in Section 6.

# 2 APPROXIMATE BAYESIAN INFERENCE FOR DEEP LEARNING

The Bayesian neural networks considered in this work were chosen to encompass a broad range of posterior approximations. While HMC provides asymptotically exact samples from the posterior, VI makes local approximations to the posterior for all the weights of the network and LLA makes local approximations only for the last layer weights. We also consider other cheaper approximations which are commonly implemented such as MC Dropout and Deep Ensembles.

## 2.1 HAMILTONIAN MONTE CARLO

The first application of MCMC to neural networks was proposed by Neal and Hinton [1998], who introduced Hamiltonian Monte Carlo (HMC) from quantum chromodynamics to the general statistics literature. However, it wasn't until Welling and Teh [2011] introduced Stochastic Gradient Langevin Dynamics (SGLD), that MCMC for neural networks became feasible for large datasets. More recently, Cobb and Jalaian [2021] have revisited HMC and proposed novel data splitting techniques to make it work with large datasets. We use the HMC algorithm in our work.

HMC simulates the path of a particle traversing the negative posterior density space using Hamiltonian dynamics [Neal et al., 2011, Betancourt, 2017, Hogg and Foreman-Mackey, 2018]. To apply HMC to deep learning, the neural network parameter space is augmented by specifying an additional momentum variable, $m$, for each parameter, $\theta$. Therefore, for a $d$-dimensional parameter space, the augmented parameter space contains $2d$ dimensions. We can then define a log joint density as follows:

$$\log[p(\theta, m)] = \log[p(\theta|D)p(m)].\quad(1)$$

Hamiltonian dynamics allows us to travel on the contours defined by the joint density of the position and momentum variables. The Hamiltonian function is given by:

$$H(\theta, m) = U(\theta) + K(m) = constant,\quad(2)$$

where $U(\theta)$ is the potential energy and $K(m)$ is the kinetic energy. The potential energy is defined to be the negative log posterior probability and the kinetic energy is usually assumed to be quadratic in nature and of the form $K(m) = (1/2)\, m^T M^{-1} m$, where $M$ is a positive-definite mass matrix. This corresponds to the negative probability density of a zero-mean Gaussian, $p(m) = \mathcal{N}(m|0, M)$, with covariance matrix, $M$, which is usually assumed to be the identity matrix.

The partial derivatives of the Hamiltonian describe how the system evolves with time. In order to solve the partial differential equations using computers, we need to discretise the time, $t$, of the dynamical simulation using a step-size, $\epsilon$. The state of the system can then be computed iteratively at times $\epsilon$, $2\epsilon$, $3\epsilon$... and so on, starting at time zero upto a specified number of steps, $L$. The leapfrog integrator is used to solve the system of partial differential equations. Two hyperparameters, the step-size, $\epsilon$, and the number of leapfrog steps, $L$, together determine the trajectory length of the simulation. The partial derivative of the potential energy with respect to the position, $\partial U/\partial\theta$, can be calculated using the automatic differentiation capabilities of most standard neural network libraries.

In each iteration of the HMC algorithm, new momentum values are sampled from Gaussian distributions, followed by simulating the trajectory of the particles according to Hamiltonian dynamics for $L$ steps using the leapfrog integrator with step-size $\epsilon$. At the end of the trajectory, the final position and momentum variables, $(\theta^*, m^*)$, are accepted based on a Metropolis-Hastings accept/reject criterion that evaluates the Hamiltonian for the proposed parameters and the previous parameters.

## 2.2 VARIATIONAL INFERENCE

Variational inference (VI) assumes an approximate posterior from a family of tractable distributions, and converts the inference problem into an optimisation problem [Graves, 2011, Blundell et al., 2015, Blei et al., 2016]. The model learns the parameters of the distributions by minimising an Evidence Lower Bound objective (ELBO) function, which is composed of a data likelihood cost and a complexity cost that quantifies the difference between the prior and the variational approximation using KL divergence.

## 2.3 LAST-LAYER LAPLACE APPROXIMATION

Last-layer Laplace approximation (LLA) constructs Gaussian approximations around the maximum a posteriori (MAP) values learned by standard NN training using the second order partial derivatives of the loss function, $\mathcal{L}$ [Daxberger et al., 2021]. This method allows one to learn posteriors for the last layer weights of the network, $\theta^{(L)}$, while keeping the rest of the values fixed at their MAP estimates. The covariance matrix for the last layer is calculated using the empirical Fisher approximation to the Hessian, which contains information about the local curvature of the loss function for each parameter. The method assumes a zero mean Gaussian prior, $p(\theta) = \mathcal{N}(\theta; 0, \gamma^2 I)$. The prior variance, $\gamma^2$, is estimated using marginal likelihood maximisation [Immer et al., 2021, Daxberger et al., 2021].

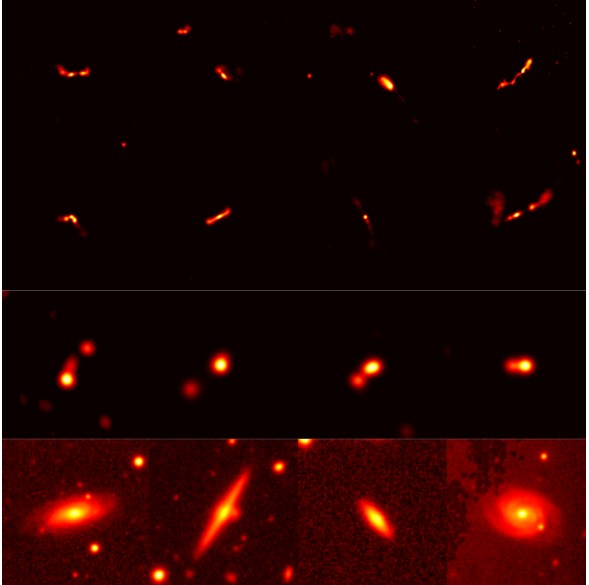

Figure 1: Images from the datasets used in this work: top two rows contain images of Fanaroff Riley Type I (FRI) and Type II (FRII) radio galaxies from the MiraBest Confident dataset on which our BNNs are trained on. The third row contains FRI/FRII galaxies from the MIGHTEE dataset. The fourth row contains optical galaxies from the GalaxyMNIST dataset. We use the MIGHTEE and GalaxyMNIST datasets to evaluate our models' ability to detect different types of distribution shifts. See Section 3 for details about the datasets.

## 2.4 MONTE CARLO DROPOUT

Another easily implemented Bayesian approximation is MC Dropout, which learns a distribution over the network outputs by setting randomly selected weights of the network to zero with probability, $p$ [Gal and Ghahramani, 2015]. MC dropout can be considered an approximation to VI, where the variational approximation is a Bernoulli distribution. Although a convenient technique, this method lacks flexibility and does not fully capture the uncertainty in model predictions, especially under covariate shift where the data distributions at training and test time are not identically distributed [Chan et al., 2020].

## 2.5 DEEP ENSEMBLES

One can use the output of multiple randomly initialised models to form a uniformly-weighted mixture model whose predictions can be combined to form an ensemble [Lakshminarayanan et al., 2017].

## 3 DATA

Radio galaxies are characterised by large scale jets and lobes which can extend up to mega-parsec distances from the central black hole and are observed in the radio spectrum. The original binary classification scheme proposed to classify such extended radio sources was based on the ratio of the extent of the highest surface brightness regions to the total extent of the galaxy, Fanaroff and Riley [1974]. FRI galaxies are edge-darkened whereas FRII galaxies are edge-brightened. Over the years, several other morphologies such as bent-tail [Rudnick and Owen, 1976, O'Dea and Owen, 1985], hybrid [Gopal-Krishna and Wiita, 2000], and double-double [Schoenmakers et al., 2000] sources have also been observed and there is still a continuing debate about the exact interplay between extrinsic effects, such as the interaction between the jet and the environment, and intrinsic effects, such as differences in central engines and accretion modes, that give rise to the different morphologies.

We train our BNNs on the MiraBest Confident dataset [Section 3.1] and use the MIGHTEE [Section 3.2] and GalaxyMNIST [Section 3.3] datasets to test the ability of our BNNs to detect different types of distribution shifts.

## 3.1 MIRABEST

The MiraBest dataset used in this work consists of 1256 images of radio galaxies of $150 \times 150$ pixels pre-processed to be used specifically for deep learning tasks [Porter and Scaife, 2023]. The galaxies are labelled using the FRI and FRII morphological types based on the definition of [Fanaroff and Riley, 1974] and further divided into their subtypes. In addition to labelling the sources as FRI, FRII and their subtypes, each source is also flagged as 'Confident' or 'Uncertain' to indicate the human classifiers' confidence while labelling the dataset. In this work we use the MiraBest Confident subset and consider only the binary FRI/FRII classification during training, see Figure 1 (top two rows) for some examples. The training and validation sets are created by splitting the predefined training data into a ratio of 80:20. The final split consists of 584 training samples, 145 validation samples, and 104 withheld test samples.

The MiraBest dataset was constructed using the sample selection and classification described in Miraghaei and Best [2017], who made use of the parent galaxy sample from Best and Heckman [2012]. Optical data from data release 7 of Sloan Digital Sky Survey [SDSS DR7; Abazajian et al., 2009] was cross-matched with NRAO VLA Sky Survey [NVSS; Condon et al., 1998] and Faint Images of the Radio Sky at Twenty-Centimeters [FIRST; Becker et al., 1995] radio surveys.

## 3.2 MIGHTEE

The MIGHTEE dataset is constructed using the Early Science data products from the MeerKAT International GHz Tiered Extragalactic Exploration survey [MIGHTEE; Heywood et al., 2022]. MIGHTEE is an ongoing radio continuum survey being conducted using the MeerKAT telescope, which is one of the precursors to the Square Kilometer Array (SKA). The survey provides radio continuum, spectral line and polarisation data, of which we use the radio continuum data and extract images for the COSMOS and XMMLSS fields. While there are thousands of objects in these fields, expert labels are only available for 117 objects. We use the data pre-processing and expert labels made available by Slijepcevic et al. [2024]. The dataset contains classifications based on the consensus of five expert radio astronomers. The final sample contains 45 FRI and 72 FRII galaxies, see Figure 1 (third row). We note that the MIGHTEE dataset contains significant observational differences from the MiraBest dataset.

## 3.3 GALAXY MNIST

In addition to considering different datasets of radio galaxies which have been curated using data from radio telescopes, we also evaluate our models on data collected from optical telescopes. Optical images of galaxies contain different features and in a sense represent completely out-of-distribution galaxies which well-calibrated models should classify with a very high degree of uncertainty so that they can be flagged for inspection by an expert.

We use the GalaxyMNIST[1] dataset which contain images of $10,000$ optical galaxies classified into four morphological types using labels collected by the Galaxy Zoo citizen science project, see Figure 1 (last row) for examples. The galaxies are drawn from the Galaxy Zoo Decals catalogue [Walmsley et al., 2022]. We resize the high resolution images from 224x224 to 150x150 to match the input dimensions of our model. We construct a small test set of $104$ galaxies from the dataset to evaluate the out-of-distribution detection ability of our BNNs.

## 4 EXPERIMENTS

Code for the experiments conducted in this work is available at: `https://github.com/devinamhn/RadioGalaxies-BNNs`.

## 4.1 MODEL ARCHITECTURE

We use an expanded LeNet-5 architecture with two additional convolutional layers with 26 and 32 channels, respectively, to be consistent with the literature on using BNNs for classifying the MiraBest dataset [Mohan et al., 2022]. The model has $232,444$ parameters in total.

## 4.2 HMC INFERENCE

We use the HAMILTORCH package[2] developed by Cobb and Jalaian [2021] for scaling HMC to large datasets. Using their HMC sampler, we set up two HMC chains of $200,000$ steps using different random seeds and run it on the MiraBest Confident dataset. We use a step size of $\epsilon = 10^{-4}$ and set the number of leapfrog steps to $L = 50$. We specify a Gaussian prior over the network parameters and evaluate different prior widths, $\sigma = \{1, 10^{-1}, 10^{-2}, 10^{-3}\}$, using the validation data set. We find that $\sigma = 10^{-1}$ results in the best predictive performance and consequently use it to define the prior width for all weights and biases of the neural network in our experiments. To compute the final posteriors we thin the chains by a factor of $1000$ to reduce the autocorrelation in the samples and obtain 200 samples. A compute time of 170 hrs is required to run the inference on two Nvidia A100 GPUs. The acceptance rate of the proposed samples is $97.62\%$. We repeat the inference with data augmentation in the form of random rotations.

**Assessing Convergence**: The Gelman-Rubin diagnostic, $\hat{R}$, is used to assess the convergence of our HMC chains [Gelman and Rubin, 1992]. If $\hat{R} \approx 1$ we consider the HMC chains for that particular parameter to have converged. We examine the convergence of the last layer weights and find that using data augmentation leads to a higher proportion of weights with $\hat{R} \geq 1$. We also monitor the negative log-likelihood and accuracy, which converge by the $100,000^{\text{th}}$ inference step.

## 4.3 OTHER INFERENCE METHODS

We conduct 10 experimental runs for each inference method presented in this section using different random seeds and random shuffling of data points between the training and validation datasets.

### 4.3.1 Deep Ensembles

We train 10 non-Bayesian CNN models with different random seeds and randomly shuffled training:validation splits to construct the posterior predictive distribution by combining the softmax values obtained for each galaxy in our test set. The models are trained for 600 epochs using the

---

[1]`https://github.com/mwalmsley/galaxy_mnist`

[2]`https://github.com/AdamCobb/hamiltorch`

Adam optimiser with a learning rate of $10^{-4}$ and weight decay $10^{-6}$. We use a learning rate scheduler which reduces the learning rate by 10% if the validation loss does not improve for two consecutive epochs and use an early stopping criterion based on the validation loss.

### 4.3.2 MC Dropout

A dropout rate of 50% is implemented before the last two fully-connected layers of our neural network. This dropout configuration performed better compared to implementing dropout only before the last layer of the network. The network is trained for 600 epochs using the Adam optimser with a learning rate of $10^{-3}$ and a weight decay of $10^{-4}$. We use a learning rate scheduler which reduces the learning rate by 10% if the validation loss does not improve for two consecutive epochs and use an early stopping criterion based on the validation loss.

### 4.3.3 LLA

We use the MAP values learned by our non-Bayesian CNNs to construct our last-layer Laplace approximation using the LAPLACE package[3] developed by Daxberger et al. [2021]. We use a diagonal factorisation of the Hessian. The optimised prior standard deviation found using marginal likelihood maximisation for 10 experimental runs lies between $\sigma \in [0.03, 0.04]$.

### 4.3.4 VI

We make a Gaussian variational approximation to the posterior and find that our model is optimised with a Gaussian prior width $\sigma = 0.01$. We also test a Laplace prior following [Mohan et al., 2022], but find that it does not lead to a significant performance improvement. Results are reported for a tempered VI posterior, with $T = 0.01$ (see note below). The network is trained for 1500 epochs using the Adam optimser with a learning rate of $5.10^{-5}$. A compute time of 40 mins is required to train the VI model on a single Nvidia A100 GPU.

**Note: Data augmentation and the cold posterior effect** Several published works have reported that their BNNs experience a "cold posterior effect (CPE)", according to which the posterior needs to be down-weighted or tempered with a temperature term, $T \leq 1$, in order to get good predictive performance [Wenzel et al., 2020]:

$$P(\theta|D) \propto (P(D|\theta)P(\theta))^{1/T}. \qquad (3)$$

Previous work on using VI for radio galaxy classification has shown that the "cold posterior effect" (CPE) persists

---

[3] https://github.com/AlexImmer/Laplace

even when the learning strategy is modified to compensate for model misspecification with a second order PAC-Bayes bound to improve the generalisation performance of the network [Mohan et al., 2022, Masegosa, 2019]. We do not observe a CPE when we use samples from our HMC inference to construct the posterior predictive distribution for classifying the MiraBest dataset. However, the effect still persists in our VI models. In the general Bayesian DL literature, some authors argue that CPE is mainly an artifact of data augmentation [Izmailov et al., 2021], while others have shown that data augmentation is a sufficient but not necessary condition for CPE to be present [Noci et al., 2021]. We find that data augmentation does not have a significant effect on the cold posterior effect observed in our VI models. However, it does lead to a different degree of trade-off between test error and uncertainty calibration error for our HMC model. The effect of augmentation on performance is further discussed in Section 5.

## 5 EVALUATION

To construct the posterior predictive distributions for a single experimental run of VI, LLA and MC Dropout, we obtain $N = 200$ samples from their posterior distributions and calculate $N$ Softmax probabilities for each class, for each galaxy in our test set. For Deep Ensembles we use $N = 10$ samples. In case of HMC we use the 200 samples obtained after thinning the chains for evaluation.

### 5.1 PREDICTIVE PERFORMANCE

We use the expected value of the posterior predictive distribution to obtain the classification of each galaxy in the MiraBest Confident test set and calculate the test error for a single experimental run by taking an average of the classification error over the entire test set. We report the mean and standard deviation of the test error for 10 experimental runs, see Table 1.

VI has the best predictive performance, irrespective of whether data augmentation is used or not. The low standard deviation values for VI indicate that the mean of the posterior predictive distribution found by VI optimisation is robust to random seeds and shuffling. The same does not hold true for LLA and Dropout, which are the two worst performing models. Deep ensembles lie somewhere in between. The MAP value reported in Table 1 is chosen on the basis of the lowest validation loss from the ensemble of CNNs that we trained.

### 5.2 UNCERTAINTY CALIBRATION

We report the expected uncertainty calibration error [UCE; Gal and Ghahramani, 2015, Laves et al., 2019, Mohan et al.,

Table 1: Test error and uncertainty calibration error (UCE) of the predictive entropy for all the Bayesian neural networks considered in this work. We also provide a baseline MAP error percentage. Inference methods with a (*) indicate that no data augmentation was used during inference for those experiments. See Sections 5.1 and 5.2.

| Inference | Error (%) $\downarrow$ | UCE $\downarrow$ |
|---|---|---|
| HMC | $4.16 \pm 0.45$ | $14.76 \pm 0.95$ |
| HMC* | $6.24 \pm 0.45$ | $12.65 \pm 0.01$ |
| VI | $3.94 \pm 0.01$ | $12.77 \pm 6.11$ |
| VI* | $3.84 \pm 0.01$ | $12.32 \pm 6.36$ |
| LLA | $8.85 \pm 2.09$ | $23.84 \pm 3.54$ |
| Dropout | $7.88 \pm 2.81$ | $25.75 \pm 4.44$ |
| Ensembles | $7.69 \pm 0.27$ | $24.41$ |
| MAP | $5.76$ | |

2022] of the predictive entropy for our posterior distributions in Table 1. For HMC, VI, LLA and MC Dropout, we use the $64\%$ credible intervals of the posterior predictive distributions to calculate UCE. For Deep Ensembles, we use the entire posterior predictive distribution constructed using the 10 ensemble members.

We find that HMC without data augmentation is the most well-calibrated BNN for the radio galaxy classification problem. HMC with data augmentation has a higher UCE. VI models with and without data augmentation are similarly calibrated. The high standard deviation values show how sensitive VI is to initialisation, and this is a well documented issue in the literature [Altosaar et al., 2018, Rossi et al., 2019]. LLA, MC Dropout and Deep Ensembles are very poorly calibrated compared to HMC and VI.

We refrain from reporting the mutual information and conditional entropy as measures of epistemic and aleatoric uncertainty since they are known to be dependent on model specification and class separability [Hüllermeier and Waegeman, 2021], making them difficult to interpret given our small statistical sample of radio galaxies. More recently, [Wimmer et al., 2023] have also shown that the additive decomposition of total predictive uncertainty into mutual information and conditional entropy breaks down in machine learning settings where we have access to a limited number of data samples. They suggest that the difference between predictive entropy and mutual information can at most be interpreted as a lower bound on the aleatoric uncertainty, which converges to the true value when the model learns the true data generating distribution.

## 5.3 DETECTING DISTRIBUTION SHIFT

When neural networks are deployed in real-world applications, the independent and identically distributed (i.i.d.) assumption often breaks down and leads to different types of distribution shifts. According to the i.i.d. assumption, the training and test sets are drawn from the same joint distribution defined by the input data and their labels

$(x, y) \sim P(X, Y)$. Covariate shift occurs when there is a change in the input data distribution, $P(X)$, but no shift in the distribution of labels, $P(Y)$, at test time. This can be due to domain shift, for example when the model is faced with galaxies from a new telescope facility. Another type of shift occurs when the distribution of labels, $P(Y)$, changes at test time due to the presence of new classes. This is known as semantic shift. Some degree of semantic shift is expected when telescopes with improved resolution reveal new morphologies of galaxies.

In order to evaluate the sensitivity of our BNNs to different types of distribution shifts, we need a scoring function which can distinguish between in-distribution (iD) and distribution-shifted test galaxies without adding significant computational overhead. Liu et al. [2020] showed that energy scores provide an easy to implement post-hoc scoring mechanism for discriminative classification models.

We calculate energy scores for different test samples, $x$, for all the datasets described in Section 3 using the logit values, $f_i(x)$, for each class, $i$, following Liu et al. [2020]. For a non-Bayesian model, an input sample, $x$, is mapped to a scalar energy value as follows:

$$\mathrm{E}(x; f) = -T.\log \sum_{i}^{K} e^{f_i(x)/T}, \qquad (4)$$

where the temperature term, $T$, is set to 1. For our Bayesian models we calculate the average energy value per input sample using $N$ posterior samples:

$$\tilde{\mathrm{E}}(x; f) = \frac{1}{N} \sum_{j}^{N} -T.\log \sum_{i}^{K} e^{f_i(x)/T}. \qquad (5)$$

In this framework, out-of-distribution (OoD) samples are expected to have higher energy.

Histograms of energy values for the different inference methods considered in this work are shown in Figure 2. We use the models with the lowest validation error from the experiments to calculate the energy scores. We see that the iD

MiraBest Confident samples get mapped to a larger interval of energy values by our HMC and VI models. In comparison, the energy scores for iD samples lie in a very narrow interval for Deep Ensembles, MC Dropout and LLA, which suggests that fewer iD samples have been pushed to lower energy values.

We find that HMC and VI models are good at separating the OoD optical galaxies from the GalaxyMNIST dataset, see Figure 2a and Figure 2b. For all other models, there is a significant degree of overlap between the iD and OoD samples, see Figure 2.

The FRI/FRII galaxies from the MIGHTEE dataset present a significant dataset shift due to differences in observational properties. MIGHTEE galaxies get mapped to a large interval of energy values, in some cases extending upto $E = -90$. However, HMC is the only model for which there exists a clear distinction between iD FRI/FRII galaxies from MiraBest Confident and distribution-shifted FRI/FRII galaxies from MIGHTEE. We also note that LLA maps some of the MIGHTEE galaxies to energies higher than OoD GalaxyMNIST data, see Figure 2c.

# 6  DISCUSSION

A certain degree of trade-off exists between a model's predictive performance and calibration. While VI has the best predictive performance, HMC without data augmentation is the most well-calibrated model and only $2.5\%$ less accurate. HMC with data augmentation has a better predictive performance, but is less calibrated than HMC without data augmentation. A similar trend has also been reported by Krishnan and Tickoo [2020], who propose a loss function which optimises for both accuracy and calibration.

The differences in dataset separability via energy scores for different BNNs can be better understood if we examine the way in which each of these models is being optimised. LeCun et al. [2006] show that many modern learning algorithms can be interpreted as energy-based models. In the energy-based framework, different loss objectives cause certain inputs' energies to be pulled up/down. LLA, Deep ensembles and MC Dropout are all trained by minimising the negative log likelihood (NLL) loss plus some regularisation term due to weight decay. Our evaluation suggests that NLL training is not be able to shape the energy functional well enough to distinguish between the datasets we have considered. While HMC is directly sampling from an energy surface that is proportional to the log of the posterior distribution, is case of VI the ELBO provides a well optimised surrogate energy function. Our HMC and VI models seem to have learned good energy surfaces. LeCun et al. [2006] also note that softmax probabilities can be considered good if the energy function is estimated well enough from the data. Perhaps this is also why HMC and VI are the better

calibrated models among all those we have considered in this work.

Our observations on the cold posterior effect (CPE) contradict the results presented in Izmailov et al. [2021]. They suggest that the CPE is largely due to data augmentation. While our HMC model does not require any tempering, the VI models require temperatures below $T = 0.01$ to produce good predictive performance. We also found that data augmentation does not have a significant impact on the CPE observed in our models. Finding the cause of the cold posterior effect observed in VI for radio galaxy classification is still an open research question. Thus we find that results from the CS literature where models are trained on terrestrial datasets often do not translate to domain-specific applications. While Deep Ensembles are generally considered a good approximation to the Bayesian posterior, [Seligmann et al., 2023] recently showed that single-mode BDL algorithms approximate the posterior better than Deep Ensembles. We also find that Deep Ensembles do not work as well as VI and HMC for our application.

Through this work, we have identified VI as the most promising method for our application given the computational cost of HMC. In future work, we plan to develop and improve our VI implementation further by using alternate optimisation strategies based on natural gradient descent [Shen et al., 2024, Khan and Rue, 2021] and proximal gradient descent [Kim et al., 2023]. We also plan to investigate the cold posterior effect further, both experimentally and theoretically. To do this we will examine the effect of data curation, which requires the creation of new datasets. Additionally, to examine robustness to prior misspecification, we plan to develop different divergence metrics for the ELBO cost function. Future work could also develop BNNs for self-supervised learning to exploit larger unlabelled datasets in astronomy.

# 7  CONCLUSIONS

In this work we have evaluated different Bayesian neural networks for the classification of radio galaxies. We found that Hamiltonian Monte Carlo and variational inference perform well at our model and dataset scales for the three criteria we considered: predictive performance, uncertainty calibration and ability to detect distribution shift. Commonly used Bayesian NNs such as MC Dropout and Deep Ensembles are poorly calibrated for our application. Since HMC is very computationally heavy, optimising VI for future radio surveys might be the way forward.

**Acknowledgements**

AMS gratefully acknowledges support from an Alan Turing Institute AI Fellowship EP/V030302/1.

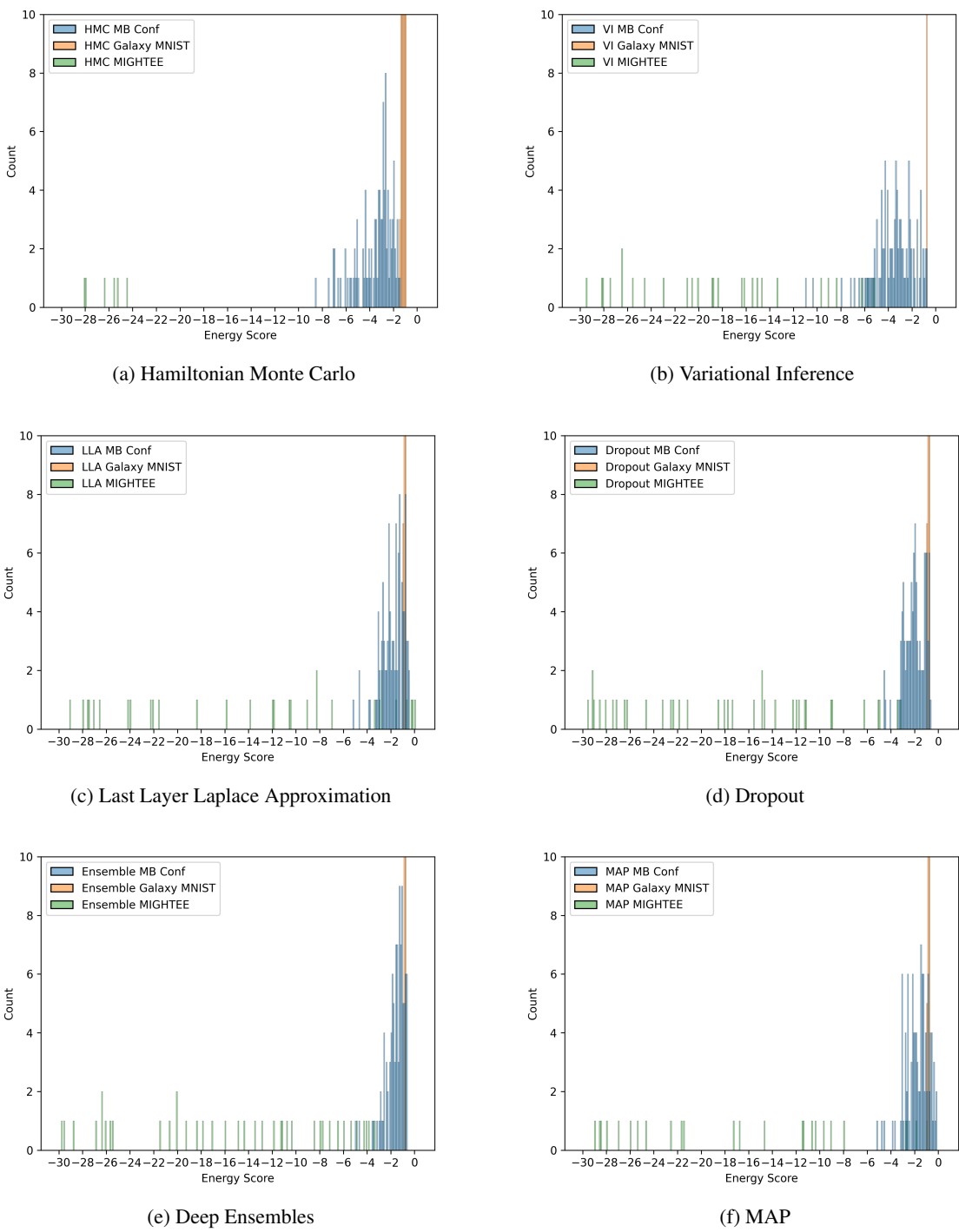

Figure 2: Detecting distribution shift with energy scores: Histograms of energy scores calculated for the MiraBest Confident (MBConf; blue), GalaxyMNIST (orange) and MIGHTEE (green) test datasets for the different models considered in this work, see Section 5.3 for details. The histograms are plotted with a bin width of 0.1. Axes are truncated so that we can examine where samples from each dataset lie. We find that HMC is the only inference method for which all the datasets can be easily distinguished.

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
