# OpenReview forum: "Evaluating Bayesian deep learning for radio galaxy classification"
_auai.org/UAI/2024/Conference — UAI 2024 poster_

### Official Review · Reviewer_PBcw · 2024-02-27

**Q2-1 Originality-Novelty:** 3
**Q2-2 Correctness-Technical Quality:** 2
**Q2-5 Clarity Of Writing:** 2

**Q1 Summary And Contributions:**

Summary:

The author evaluated Bayeisan deep learning for radio galaxy classification. Experiments of hamiltonian monte carlo, variational inference, last-layer laplace approximation, monte carlo  dropout, deep ensembles methods substantiated the performance of the proposed model.

**Q2-3 Extent To Which Claims Are Supported By Evidence:**

2: Fair: the main claims are somewhat supported by evidence (but the experimental evaluation may be weak, or does not match entirely with the claims, important baselines may be missing, proofs contain important ideas but lack rigor, algorithmic details are only discussed superficially, references are imprecise, assumptions are not sufficiently motivated or explicated, etc.).

**Q2-4 Reproducibility:**

2: Fair: key resources (e.g. proofs, code, data) are unavailable but key details (e.g. proof sketches, experimental setup) are sufficiently well-described for an expert to confidently reproduce the main results.

**Q3 Main Strengths:**

This paper is novel.

√The objectives, rationale and strengths of the study are clearly stated. The proposed model provided a principled way to model uncertainty in the predictions made by DNN models and played an important role in extracting well-calibrated uncertainty estimates on their outputs.

√The application and theory is reported in sufficient detail to allow for its replicability and reproducibility. The partition of the training,validation and test set is mentioned in the paper.  The open source codes are encouraged.

**Q4 Main Weakness:**

×The authors didn’t state the limitations of their methods.

×The manuscript can benefit from further proofreading.

**Q5 Detailed Comments To The Authors:**

1.In the section 3.3 GALAXY MNIST, there are 2 “using” in the first sentence.

**Q9 Complying With Reviewing Instructions:**

Yes

---

> ### Author Rebuttal · Authors · 2024-04-02
>
> We thank the reviewer for their comments and feedback. We are pleased that they found our paper novel and that they are satisfied with the manuscript in terms of reproducibility.
>
> We thank the reviewer for pointing out the typo, which has now been corrected. We have further proofread the manuscript since submission and corrected any other typos we could identify.
>
> We believe we have demonstrated some limitations of the methods used through our analysis. We will add text to explicitly note other limitations including the following:
> The methods that perform well are computationally heavy (e.g. HMC), so the results will not scale well with increased dataset size. Cheaper approximations to the posterior do not perform well. Therefore, there is a need to develop methods like VI to balance the tradeoff between performance and computational cost.
>
> Re: Q2-5 (Clarity of writing)- If the reviewers have specific suggestions to improve the presentation of our material, we would be happy to incorporate those changes to make our work accessible to a broader audience (including both ML researchers and astrophysicists).

---

### Official Review · Reviewer_tU4x · 2024-03-19

**Q2-1 Originality-Novelty:** 1
**Q2-2 Correctness-Technical Quality:** 3
**Q2-5 Clarity Of Writing:** 3

**Q1 Summary And Contributions:**

This paper evaluates the performance of Bayesian neural networks on radio galaxy classification problems. Various inference methods are compared, including HMC, Variational inference, Monte Carlo dropout, and deep ensembles.

**Q2-3 Extent To Which Claims Are Supported By Evidence:**

3: Good: the main claims are supported by convincing evidence (in the form of adequate experimental evaluation, proofs, (pseudo-)code, references, assumptions).

**Q2-4 Reproducibility:**

3: Good: key resources (e.g. proofs, code, data) are available and key details (e.g. proofs, experimental setup) are sufficiently well-described for competent researchers to confidently reproduce the main results.

**Q3 Main Strengths:**

The paper is well-written. The problem is a meaningful and interesting problem. The experimental results give some insights or guidelines for applying BNNs and their various inference methods to the radio galaxy classification problem and maybe other problems as well.

**Q4 Main Weakness:**

The contribution is weak. There is no new methodology proposed in this paper. Most of content of this paper, like various of inference methods are just plain description of existing works.

**Q5 Detailed Comments To The Authors:**

The authors focus on an interesting problem and have found the benefit of existing works. It is expected to propose some new idea or design to further improve the performance based on the current results.

**Q9 Complying With Reviewing Instructions:**

Yes

---

> ### Author Rebuttal · Authors · 2024-04-02
>
> We thank the reviewer for their comments and feedback. We are pleased that they found our problem meaningful and the paper well-written. We agree with the reviewer that no novel methodological contributions have been made by our work. However, the main contribution of our work is to present a novel real-world application of Bayesian NNs to radio astronomy (as also noted by Reviewer 4vmH), which will be useful in developing the data analysis pipelines for future radio astronomy surveys (the UAI call for papers invites papers on novel applications as well as theory and methodology).
> In order to make progress in developing novel methods we first need to understand and evaluate the performance of existing methods along the axes that are relevant to the scientific application of BNNs.
>
> The goal of this work is to evaluate existing Bayesian NNs and establish a baseline which will help us design new algorithms by understanding the strengths and weaknesses of the BNNs that exist in the literature. This step is necessary because we have found that results from the CS literature do not directly translate to domain-specific applications (for e.g., the cold posterior effect in our variational inference models for radio galaxy classification cannot be explained by the existing theories in the literature). This is also noted in our Discussion section. It is our hope that the experimental evaluation from our work will be useful for both astronomers as well as the scientific community in developing domain-specific Bayesian neural networks.
>
> Through this work, we have identified variational inference as the most promising method for our application and plan to develop it further in future work. We will include specific research directions we plan to pursue in our future work in the Discussion section (see response to Reviewer 4vmH for details).
>
> Some of the content of the paper, like the reviewer points out, are descriptions of existing methods. We find this necessary to give enough background to our audience which is multidisciplinary (ML researchers, astronomers and researchers working in ML for astronomy).
>
> Re: Q2-5 (Clarity of writing)- If the reviewers have any specific suggestions to improve the presentation of our material, we would be happy to incorporate those changes to make our work accessible to a broader audience (including both ML researchers and astrophysicists).

---

### Official Review · Reviewer_KfRE · 2024-03-22

**Q2-1 Originality-Novelty:** 2
**Q2-2 Correctness-Technical Quality:** 3
**Q2-5 Clarity Of Writing:** 3

**Q1 Summary And Contributions:**

This paper evaluated different Bayesian neural networks for the classification of radio galaxies.

**Q2-3 Extent To Which Claims Are Supported By Evidence:**

3: Good: the main claims are supported by convincing evidence (in the form of adequate experimental evaluation, proofs, (pseudo-)code, references, assumptions).

**Q2-4 Reproducibility:**

3: Good: key resources (e.g. proofs, code, data) are available and key details (e.g. proofs, experimental setup) are sufficiently well-described for competent researchers to confidently reproduce the main results.

**Q3 Main Strengths:**

1) Played with many real datasets
2) Tried many different models which makes a lot of combinatorial tests
3) The final discussion and findings are clear

**Q4 Main Weakness:**

No methodology contributions, it is just an experimental survey paper

**Q5 Detailed Comments To The Authors:**

No issues I want to raise in this work.

**Q9 Complying With Reviewing Instructions:**

Yes

---

> ### Author Rebuttal · Authors · 2024-04-02
>
> We thank the reviewer for their comments and positive feedback. We are pleased that they found our experiments convincing and our findings and discussions clear.
>
> Re: Q2-5 (Clarity of writing)- If the reviewers have specific suggestions to improve the presentation of our material, we would be happy to incorporate those changes to make our work accessible to a broader audience (including both ML researchers and astrophysicists).

---

### Official Review · Reviewer_4vmH · 2024-03-22

**Q2-1 Originality-Novelty:** 3
**Q2-2 Correctness-Technical Quality:** 3
**Q2-5 Clarity Of Writing:** 3

**Q10 Ethical Concerns:**

No.

**Q1 Summary And Contributions:**

The article is to evaluate the performance of Bayesian neural networks (BNNs) in radio galaxy classification problems, particularly in predictive performance, uncertainty calibration, and distribution shift detection. Authors compared the performance of different BNNs in radio galaxy morphology classification tasks, including Hamilton Monte Carlo (HMC), Variational Inference (VI), Last Laplace Approximation (LLA), Monte Carlo Dropout, and Deep Ensemble methods.Studied the sensitivity of these methods in dealing with distribution shifts in datasets, especially for radio galaxy image datasets from different radio telescopes. It was found that HMC and VI perform well in predictive performance and uncertainty calibration, while commonly used BNNs such as Dropout and deep integration have poor calibration in these applications. It is proposed that optimizing VI may be a direction forward for future radio observations, given the high computational cost of HMC.

**Q2-3 Extent To Which Claims Are Supported By Evidence:**

2: Fair: the main claims are somewhat supported by evidence (but the experimental evaluation may be weak, or does not match entirely with the claims, important baselines may be missing, proofs contain important ideas but lack rigor, algorithmic details are only discussed superficially, references are imprecise, assumptions are not sufficiently motivated or explicated, etc.).

**Q2-4 Reproducibility:**

2: Fair: key resources (e.g. proofs, code, data) are unavailable but key details (e.g. proof sketches, experimental setup) are sufficiently well-described for an expert to confidently reproduce the main results.

**Q3 Main Strengths:**

This study applies Bayesian deep learning techniques in the field of radio astronomy, which is a relatively new research direction. Especially, applying BNNs to radio galaxy classification problems and evaluating their ability in predictive performance.The study utilized multiple different BNNs methods and rigorously compared and evaluated them. In addition, the author used appropriate statistical methods (such as Gelman Rubin diagnosis) to evaluate the convergence of MCMC chains.
The research results indicate that HMC and VI perform well in radio galaxy classification tasks, which is strongly supported by experimental data. In addition, by comparing with other methods, the author clearly demonstrates the advantages and limitations of these methods.
The author provided a detailed experimental setup and method description, which will help other researchers replicate and validate the research results.

**Q4 Main Weakness:**

The research mainly focuses on specific datasets and classification tasks, which may limit the universal applicability of its conclusions. In addition, further experimental or theoretical support may be needed for the analysis of the causes of certain phenomena, such as cold peripheral effects.
The calculation cost of HMC method is relatively high, and specific adjustments may be required for the selection of hyperparameters such as temperature adjustment in VI method, which may affect reproducibility.

**Q5 Detailed Comments To The Authors:**

Although interesting results have been achieved in the classification of radio galaxies, the author may consider extending the research to other types of astronomical data or machine learning tasks to verify the generalization ability and applicability of the proposed method.
The article mentions the use of multiple BNNs methods, but does not provide a detailed discussion on why these specific methods were chosen. The author can provide more reasons for method selection and a discussion of previous related work in the methods section, which will help readers understand the background of the research and the rationality of the methods.
Regarding the temperature adjustment in the Variational Inference (VI) model. It is suggested that the author can further study this phenomenon and provide deeper insights through experiments or theoretical analysis.
In the discussion section, the author can provide a more detailed description of the future work direction, including how to improve current methods and how to apply these methods to new datasets or problems.
Please ensure that all cited work is correctly listed and that the reference format complies with the guidelines of the journal or conference. If any important related work is omitted, it should be supplemented.

**Q9 Complying With Reviewing Instructions:**

Yes

---

> ### Author Rebuttal · Authors · 2024-04-02
>
> We thank the reviewer for their in-depth feedback. We are pleased that they found our application novel and our evaluation rigorous.
>
> We believe that by focusing on specific datasets and classification tasks, we can gain insights into domain specific and domain agnostic adaptations needed for the deployment of Bayesian deep learning models in real-world applications. We thank the reviewer for suggesting extending the research to other types of astronomical data. This is something we plan to do in the longer term.
>
> The BNN methods were chosen to encompass a whole range of posterior approximations  - from the asymptotically exact HMC samples, to Gaussian approximations to the posterior of all the weights of the network (VI), to Gaussian approximations only to the last layer weights (LLA) and other cheaper approximations which are commonly implemented such as MC Dropout and Deep Ensembles. The methods have a trade-off between computational cost and performance. We will add text at the beginning of Section 2 before the description of the methods to help readers understand the background of the research and the rationality of the methods.
>
> We agree that the computational cost of HMC is relatively high, but it is the only Bayesian neural network which can be considered a 'gold standard' for Bayesian inference because it gives asymptotically exact samples. This is why we included it as a baseline against BNNs which make gross approximations to the posterior.
>
> We plan to investigate the cold posterior effect (use of temperature term in variational inference) further in our future work, both experimentally and theoretically. To do this we will examine the effect of data curation, but this requires the creation of new datasets which will require additional time.
>
> We will include detailed descriptions of future work in the Discussion including the following points:
> (i) We plan to develop and improve VI further by using alternate optimisation strategies based on natural gradient descent and proximal gradient descent;
> (ii) Examination of the cold posterior effect with respect to data curation (as stated above);
> (iii) Additionally, to examine robustness to prior misspecification, we plan to develop different divergence metrics for the ELBO cost function;
> (iv) Developing the methods for self-supervised learning to exploit larger unlabelled datasets in astronomy (which is still unexplored in the literature)
>
> We will ensure that the reference format complies with the guidelines of the conference (we used the specified template but we will double-check the guidelines).
>
> We believe we have cited all relevant work, but if there are specific papers we might have missed, please let us know and we will add those references.
>
> Code and data are publicly available and will be linked to the submission once/if accepted. We believe we have given enough experimental details in the manuscript for researchers to confidently reproduce the main results (as also reported in Q2-4 scores of R2, R3).
>
> Re: Q2-5 (Clarity of writing)- If the reviewers have specific suggestions to improve the presentation of our material, we would be happy to incorporate those changes to make our work accessible to a broader audience (including both ML researchers and astrophysicists).

---

### Meta-Review · Area_Chair_3uFh · 2024-04-17

This paper evaluates Bayesian neural networks (BNNs) in radio galaxy classification problems. It includes a range of popular BNN methodologies including variational inference, Hamilton Monte Carlo, Laplace approximation, MC dropout, and deep ensemble. The evaluation metrics employed are diverse, covering accuracy, uncertainty calibration error, and distribution shift detection based on energy scores.

Although the paper does not introduce a new BNN method, it presents a novel application of BNNs in a compelling way, with a thorough and convincing experimental design.